# Effect of Surface Hydrophilized Plastic Waste Aggregates Made by Mixing Various Kinds of Plastic on Mechanical Properties of Mortar

**DOI:** 10.3390/ma17010247

**Published:** 2024-01-02

**Authors:** Kyung-Min Kim, Young-Keun Cho

**Affiliations:** Construction Technology Research Center, Korea Conformity Laboratories (KCL), 199, Gasan Digital 1-ro, Geumcheon-gu, Seoul 08503, Republic of Korea; young@kcl.re.kr

**Keywords:** mixed plastic aggregates, sand replacement, surface hydrophilization, mortar

## Abstract

The surface hydrophilization of mixed plastic waste aggregates (MPAs) was conducted to improve the bond between an MPA and the surrounding cement matrix using two types of coating agents: a silicone amine resin and acrylic binders. The coating agents formed a physical bond with the MPAs, and the results of contact angle measurement also revealed that the surface of MPAs was hydrophilic. The workability of a mortar mix increased by up to 1.47 times with the surface hydrophilization of MPAs. Meanwhile, the compressive and flexural strengths of mortar mixes decreased by 29~43% and 72~86%, respectively, at 28 days with the surface hydrophilization of MPAs. Namely, the surface hydrophilization of MPAs was successively conducted, and the workability of mortar mixes was improved accordingly, but the compressive and flexural strengths of mortar mixes decreased as the physical bond was partially separated from not only the MPA but also the surrounding cement matrix and the surface friction was decreased.

## 1. Introduction

As the use of plastics increases worldwide for reasons such as convenience, the amount of plastic discarded is also increasing. In 2019, global plastic production was 460 million tons. However, only 9% of plastic waste was recycled, 19% of plastic waste was incinerated, and 50% of plastic waste ended up in landfills [1]. Even 22% of plastic waste was found to be improperly disposed of, such as being incinerated in open pits or dumped on land or in aquatic environments, especially in poorer countries. As such, only part of the plastic waste is recycled, and the rest is landfilled, incinerated, or mistreated, causing serious environmental pollution problems. In Korea as well, in the case of household waste, the average daily amount of plastic waste increased from 7260 tons in 2016 to 12,822 tons in 2021 [2]. Among that, 86.6% of plastic waste discharged separately was recycled, but only 13.4% of plastic waste discharged with other types of household waste was recycled. This is because such plastic waste is mixed with various types of polymers as well as other types of household waste, increasing the cost of separation and sorting. Plastic waste discharged with other types of household waste also contains a large number of foreign substances.

One of the ways to economically mass recycle plastic waste is to use it as a material in cement-based composites such as mortar and concrete [3,4,5]. By using plastic waste as sand or gravel replacement in cement-based composites, it is possible not only to recycle plastic waste more effectively but also to reduce consumption of limited natural resources such as sand and gravel.

The researches on the use of plastic waste aggregates (PWAs) in cement-based composites have mainly been undertaken by using a PWA made by a single type of polymer such as polyethylene terephthalate (PET), polypropylene (PP), polyvinylchloride (PVC), polyethylene (PE), polycarbonate (PC), and so on [6,7,8,9,10,11,12,13]. However, limited research has investigated the effect of mixed plastic waste aggregates (MPAs) composed of several types of polymers on the properties of cement-based composites, and there is also research that has simultaneously investigated the effects of different types of PWAs on cement-based composite properties [3,14,15,16].

Jacob-Vaillancourt et al. [3] reported that the compressive strength of concrete decreased at 28 days as 5%, 10%, and 20% of sand volume was replaced with MPAs. It was also found that under the same conditions of replacing 20% of sand volume with PWAs made of one type of polymer or MPA and using an air-reducing agent of 50 mL/m^3^, the compressive strength of concrete with MPAs was smaller than that of concrete using PVC aggregates but was larger than that of concrete with PWAs made of PP, PE, or PS. Ruiz-Herrero et al. [14] used two types of PWAs, each made of PVC and PE. In the same replacement of sand volume with PWAs, the compressive strength loss rate of concrete with both PVC and PE aggregates did not necessarily get larger than that of concrete with single-material aggregates of PVC or PE, and the compressive strength loss rate of concrete with both PVC and PE aggregates was the smallest when 10% of sand volume was replaced. It was also found that from 10% of sand volume with PVC aggregates, the compressive strength loss rate of concrete was equal to or smaller than that of mortar.

It has been reported that the mechanical properties of cement-based composites using PWAs are affected by the type, shape, size, and input amount of PWAs. In particular, the compressive strength of cement-based composites with PWAs usually decreases as the amount of PWA increases, even though some researchers reported that the compressive strength of cement-based composites using PWAs slightly increased for low-level PWA replacement [8,10].

Thorneycroft et al. [8] replaced 0.64% of sand volume with virgin PP fibers, and the compressive strength resulted in an improvement of 1.5% compared to the reference mix without a PWA. The reasons why PWAs reduces the compressive strength of concrete are generally known as (1) the lower strength and stiffness of PWAs compared to natural aggregates, (2) the air voids formed around PWAs due to the hydrophobicity of PWAs, and (3) the weak bond between PWAs and the surrounding cement matrix by more air voids around PWAs compared to natural aggregates [7,11].

Accordingly, some researches [8,17,18] have been undertaken to improve the weak bond between a PWA and the surrounding cement matrix by chemical or physical surface treatment of PWAs. Naik et al. [17] chemically treated a PWA (shredded high-density polyethylene) with bleach (5% hypochlorite) solution and bleach (5% hypochlorite) solution with 4% sodium hydroxide and attempted to improve bonding between the treated PWA and the surrounding cement matrix through chemical reaction. The surface treatment with bleach solution with sodium hydroxide was reported to be more effective on preventing compressive strength loss by the PWA than the surface treatment with just the bleach solution. Thorneycroft et al. [8] similarly treated the surface of PET and PP fiber aggregates with a solution of bleach (sodium hypochlorite) with caustic soda (sodium hydroxide). The compressive strength of concrete using PET aggregates that were chemically treated was lower than that of concrete with PET aggregates that were not chemically treated. This was assumed to be because the crystals, which formed on the surface of PET aggregates by chemical treatment, dissolved and decomposed during mixing and curing of concrete. Choi et al. [18] coated the surface of PET aggregates with ground granulated blastfurnace slag and confirmed that calcium hydroxide was formed on the aggregate surface. However, the compressive strength of concrete also decreased with the increase in the volume of the coated PET aggregates.

To improve the weak bond between PWAs and the surrounding cement matrix, some researches have been conducted on chemical surface treatment of PWAs for concrete. Although it has been shown that the compressive strength of concrete has been partially improved by chemical surface treatment of PWAs, research on improving the bond of MWAs and the surrounding cement matrix in cement-based composites is still limited, including for PWAs.

The purpose of this study is to recycle plastic waste discharged with other types of household waste as sand replacement in mortar for civil structures. Therefore, the surface of the MPAs was modified to be hydrophilic to improve the bond between the MPA and the surrounding cement matrix by increasing the polar surface energy of it using two types of coating agents: a silicone amine resin and acrylic binders. Then, the effect of the surface hydrophilization of MPAs on the bond between the MPA and the surrounding cement matrix and consequently on properties of mortar was experimentally evaluated.

## 2. Materials and Methods

### 2.1. Materials

#### 2.1.1. Cement and Aggregates

An ordinary Portland cement with the chemical composition in Table 1 was employed. The specific surface area and density of the cement are 3510 cm^2^/g and 3.15 g/cm^3^, respectively. As a fine aggregate, ISO standard sand [19] with a maximum size of 2 mm was employed with the properties in Table 2.

#### 2.1.2. Mixed Plastic Waste Aggregates

MPAs were manufactured by mixing and extruding various kinds of plastic waste from household waste collected in Korea (Gyeonggi-do) without separation or sorting.

First, the collected plastic waste in 2019 was washed and cut, and 50 measurements by infrared spectroscopy were conducted to analyze the polymers that make up plastic waste. As a result, plastic waste was found to contain polymers similar to PE, including linear low-density polyethylene (LLDPE), low density polyethylene (LDPE), high density polyethylene (HDPE), PP, PET, ethylene vinyl acetate (EVA), and PVC, as shown in Table 3. The composition ratio of polymers also appeared to vary depending on the collection period of plastic waste, but PE was found to be the most common.

Figure 1 illustrates the entire process of manufacturing MPAs. The MPA manufacturing process is as follows: cutting plastic waste → washing for removal of foreign substances and dehydration → primary melting → secondary melting and extrusion → cooling. Meanwhile, in this study, we intend to ultimately apply MPAs to the concrete of heavy civil structures, and blast furnace slag (BFS) fine powder was added to increase the low density of the plastic below 1.00. The BFS fine powder was put into a melter during the secondary melting with the same volume ratio as the input plastic waste. 

Table 4 shows the physical properties of MPAs. The absorption rate of MPAs was about 1.36 times higher than that of sand due to the large number of pores inside MPAs formed by melting several types of polymers at the same time during the melting process. Moreover, the density of the MPA where BFS fine powder was added resulted as 1.38 g/cm^3^, which is higher than the density of 0.77~0.89 g/cm^3^ for the MPA and PWA, each made of LDPE and PP [20], because the BFS fine powder filled some pores formed during the primary melting process. Meanwhile, MPAs were manufactured by cutting the plastic extruding into a long rod through the extrusion process into an aggregate size. The MPA is thus rod-shaped with relatively high flatness and low elongation, as shown in Figure 2a. MPAs also have a single particle distribution with the particle size between 2.5~5 mm, differing from the typical particle size distribution of fine aggregates, as illustrated in Figure 2b.

### 2.2. Preparations

#### 2.2.1. Hydrophilization of Mixed Plastic Waste Aggregates

Figure 3 illustrates the process of the surface hydrophilization of the MPA, and it is conducted through the following order: heating → impregnation → drying. The MPA was first heated above 50 °C to activate the polymerization reaction and improve adhesion with the coating agent in the furnace. The MPA was then impregnated with the coating agent for one hour. Finally, the MPA was removed from the coating agent and dried in a laboratory environment with a temperature of 20 ± 2 °C and humidity of 60% for 24 h.

As the coating agents for the surface hydrophilization of the MPA, two types of common primers, which primarily apply to the surface of members made of inorganic materials in order to improve the bond with finishing materials made from organic materials, were used. One is a modified silicone resin, and the other is an acrylic binder.

Silicone is a polymer connected by chemical bonds such as silicon and oxygen and contains organic functional groups that chemically bond with organic materials. In this study, a liquid silicone amine resin (CA1), which is a silicone resin containing amine, a hydrophilic group, was used to decrease the surface tension and introduce hydrophilicity to the surface of MPA.

On the other hand, an acrylic binder is formed by polymerizing various acrylate monomers with acrylic or methacrylic acid esters. An acrylic binder is also hydrophilic because it contains chemically high polar carboxyl groups, which are known to be more reactive with cementitious materials [17]. In this study, two types of liquid acrylic binders, acrylic colorant (CA2) and ethyl acrylate binder (10 wt%) (CA3), were used to increase the polar surface energy and introduce hydrophilicity to the surface of MPA.

Figure 4 presents the surface condition of the unmodified MPA and surface-modified MPAs with two types of coating agents. Unlike the unmodified MPA, the surface-modified MPAs have glossy and smooth surfaces due to the coating agents. Meanwhile, in the case of the MPA modified with coating agent CA3, there were some parts on the surface that were not treated with the coating agent.

#### 2.2.2. Mortar Mix Designs

Table 5 shows the mix design of mortar. Mix designs were planned as investigating the effect of surface hydrophilization of MPA on mortar strength according to the two types of coating agents: a silicone amine resin and two types of acrylic binders. The replacement volume content of sand with MPA was set at 34%, and the water-cement ratio was fixed at 38%.

Three beams of 40 mm × 40 mm × 160 mm were casted from each mix for a strength test and cured in water at a temperature of 20 ± 2 °C for 7 and 28 days prior to the strength test [19].

### 2.3. Experimental Method

The surface hydrophilization of MPA was basically evaluated by analyzing a contact angle measurement result. A surface is considered to be hydrophilic when the contact angle is less than 90° [21]. The contact angle was measured using a contact angle measuring instrument (DSA25, KRUSS) [22]. For the contact angle, the angle between 5 μL of deionized water droplets and the MPA surface was measured six times under the conditions of 60% relative humidity and 20 °C. Additionally, the physical bond formation was evaluated by scanning electron microscopy (SEM)-energy dispersive X-ray spectroscopy (EDS) analysis.

The flow, compressive, and flexural tests were performed to evaluate the mortar properties. The flow test was performed immediately after mixing according to ASTM C1437 [23] to evaluate the workability of the mixtures using MPAs. The flexural test was performed first at a loading rate of 50 N/s using a 1000 kN UTM, and then the compressive test was performed at a loading rate of 2400 N/s with the 40 mm × 40 mm samples, which had been split into two from the flexural test [19]. For all the mixtures in Table 5, the flexural and compressive tests were conducted on three and six specimens, respectively, each at 7 and 28 days.

The bond between MPA and the surrounding cement matrix was indirectly evaluated by an optical microscope image and SEM analysis. The mortar internal cross-section condition was also evaluated by an optical microscope image.

## 3. Results and Discussion

### 3.1. Surface Hydrophilization Results

Table 6 shows the contact angles of the unmodified MPA and the surface-modified MPAs with two types of coating agents, and Figure 5 illustrates the average contact angles by six measurements for all MPA.

The contact angles of the surface-modified MPAs with the coating agents of CA1, CA2, and CA3 were, on average, 61.3°, 85.6°, and 90.8°, relatively, and that of the unmodified MPA was, on average, 107.7°. The surface of the unmodified MPA was confirmed to be hydrophobic with a contact angle greater than 90°. Meanwhile, the surfaces of MPAs modified with coating agents CA1 and CA2 were modified to be hydrophilic with contact angles less than 90°. The surface-modified MPA with CA1 was found to be most effective in modifying the surface of MPA to be hydrophilic. This is believed to be due to the low surface tension of CA1.

In the case of the MPA modified with coating agent CA3, the contact angle was slightly greater than 90°. Moreover, since the contact angle measurements were less or greater than 90°, the MPA surface is considered to be partially hydrophilic. This is related to the parts not treated with the coating agent, as shown in Figure 4d.

Figure 6 presents the SEM-EDS images of the unmodified MPA and the surface-modified MPAs with two types of coating agents.

Through the SEM images, it can be confirmed that there are coating layers on the surfaces of the surface-modified MPAs with two types of coating agents, although the thicknesses of these coating layers are not constant. In other words, the coating agents formed a physical bond in the form of a thick layer. The surface-modified MPA with CA1, which was considered to be the most effective for the surface hydrophilization of MPAs, generally formed the thickest coating layer on the surface, as shown in Figure 6 and Figure 7. On the contrary, the surface-modified MPA with CA3 with the highest contact angle formed the thinnest coating layer in the surface. For the surface-modified MPA with acrylic binder types CA2 and CA3, some voids were also found between the MPA surface and the coating layer. Therefore, it is believed that this coating layer thickness and the voids between the MPA surface and coating layer also influenced the contact angle results above in addition to the surface tension of the coating agent.

Meanwhile, as a result of analyzing the distribution of elements constituting MPAs and the coating agent at the boundary between the MPA and the coating agent through EDS analysis, it was found that most of the elements that make up the coating agents were carbon, as presented in Figure 6 and Table 7, because the coating agents were organic materials.

### 3.2. Fresh Properties of Mortar

Table 8 and Figure 8 show the flow test results. The flow decreased by 53~78% as 34% of the sand volume was replaced with MPAs. The workability of all the mixtures with MPAs was lower than that of the reference mixture, Ref. This may be because the rod shape of the MPA reduced the friction with the mortar content [24], and the packing density effect of the MPA could not be expected due to the single particle distribution of the MPA, as illustrated in Figure 2b [25].

Meanwhile, the surface hydrophilization of MPAs was found to increase the workability of mortar mixes, and this is believed to be because the coating agents make the MPA surfaces glossy and smooth, which reduces the friction of the MPA surface [26]. It was found that mix C3_MPA using the surface-modified MPA with CA3, which showed the largest contact angle, had the greatest workability, and the workability was not inversely proportional to the contact angle. 

### 3.3. Mechanical Properties of Mortar

Table 8 and Figure 9 show the compressive and flexural strength test results. The compressive strength decreased by 29~45% at 7 days and 29~44% at 28 days, as 34% of the sand volume was replaced with MPAs. This may be due to the air voids formed around MPAs and between the fine aggregates, as presented in Figure 10. However, this level of compressive strength loss due to MPA replacement is believed to not be relatively low compared to the compressive strength loss results of Ruiz-Herrero et al. [14]. Ruiz-Herrero et al. [14] found that the compressive strength loss rate of mortar where 20% of sand was replaced with PVC was over 70% and was even greater in the case of PE at 28 days. Meanwhile, the air void formed around PWAs is generally known to be due to the hydrophobicity of PWAs [7]. However, the air voids formed around the MPAs were also found in the case of the mixtures using the surface-modified MPA, as shown in Figure 10, and it is believed that water was partially absorbed into the hydrophilic modified surface, forming the air voids.

In addition, the compressive strengths of the mixtures using the surface-modified MPA were found to be lower than that of mix NC_MPA using the unmodified MPA for both 7 and 28 days. This is believed to be because the surface friction of the MPAs was reduced by the surface modification [26]. However, the mixtures C1_MPA and C2_MPA using hydrophilically modified MPA, as a result of contact angle analysis, had the same standard deviation of compressive strength at 28 days of 1.7%, while the standard deviation of compressive strength at 28 days was 2.9% in the case of mix NC_MPA. The surface modification is thus considered to be advantageous in securing the compressive strength at 28 days in terms of repeatability.

The flexural strength decreased by 49~76% at 7 days and 72~96% at 28 days as sand was replaced with an MPA. It was found that the longer the age, the less the flexural strength of the mixtures with MPAs, and the flexural strength loss due to the replacement of sand with MPAs was low compared to the compressive strength loss. This is believed to be because part of the MPA resisted the expansion of microcracks according to the flexural loading [6]. The flexural strength of the mixtures using the surface-modified MPA was lower than that of mix NC_MPA using the unmodified MPA similar to the compressive strength results. This is believed to be because the surface coating made the surface of the MPA partially smooth and reduced the microcrack expansion resistance.

### 3.4. Bond between MPA and Cement Matrix

Figure 11 shows the SEM images of the interface between the MPA and the surrounding cement matrix. In all the mixtures with MPAs, the MPA was found to debond from the surrounding cement matrix in the SEM images, in addition to the air voids forming around the MPA, as presented in Figure 10.

In the case of the mixtures using the surface-modified MPA, the separation between the MPA and the surrounding coating agent in some parts was also inspected, as well as the separation between the surface coating agent of the MPA and the surrounding cement matrix. In the case of mix C3_MPA using the surface-modified MPA with CA3, cracks occurred in the coating agent too. It seems that the bonds between the MPA and the surrounding coating agents were weakened as the coating agents partially reacted with the surrounding cement matrix during the cement hydration process.

## 4. Conclusions

In this study, the surface hydrophilization of MPAs was conducted with the aim of improving the bond between the MPA and the surrounding cement matrix. The change in the surface of the MPA and its effects on the bond between the MPA and the surrounding cement matrix and the mortar properties were evaluated. The study findings can be summarized as follows.

It was confirmed that the surface of MPAs was successfully modified to be hydrophilic through analysis of the contact angle analysis results, and the surface-modified MPA with a silicone amine resin had a greater degree of hydrophilicity than surface-modified MPA with acrylic binders due to the low surface tension. The degree of hydrophilicity of the MPA tended to increase when the thickness of the coating layer was thick and there were little voids between the MPA and the coating layer.The workability of mortar mixes using the surface-modified MPA increased as the surface friction of the MPA decreased with the coatings. Moreover, the acrylic binders were more effective in improving the workability of mortar mixes.The compressive strength of mortar mixes decreased with the surface hydrophilization of the MPA. However, the mortar mix using the surface-modified MPA showed the possibility of securing the compressive strength at 28 days in terms of repeatability. The flexural strength of the mortar mixes using the surface-modified MPA was also found to be lower than that of the mortar mix using the unmodified MPA.The surface of the MPA was modified to be hydrophilic through the surface coating of the MPA. However, the surface friction of the MPA was reduced, and water was partially in the surface due to the surface modification. The coating layer also seemed to act as another weak layer during the cement hydration process. As a result, the compressive and flexural strengths were considered to decrease.Consequently, the results of this study showed that the surface hydrophilization of the MPA was effective in improving the workability of mortar mixes using MPAs, and the strength properties of mortar mixes using the surface-modified MPA were influenced by the state of the physical bond formed by the coating agents. MPAs have a lot of potential in environmental terms, such as mass recycling of plastic waste and reducing consumption of limited natural resources. Thus, it is expected that the applicability of MPAs to cement-based composites will increase through additional research on the quality control of the physical bond.

## Figures and Tables

**Figure 1 materials-17-00247-f001:**
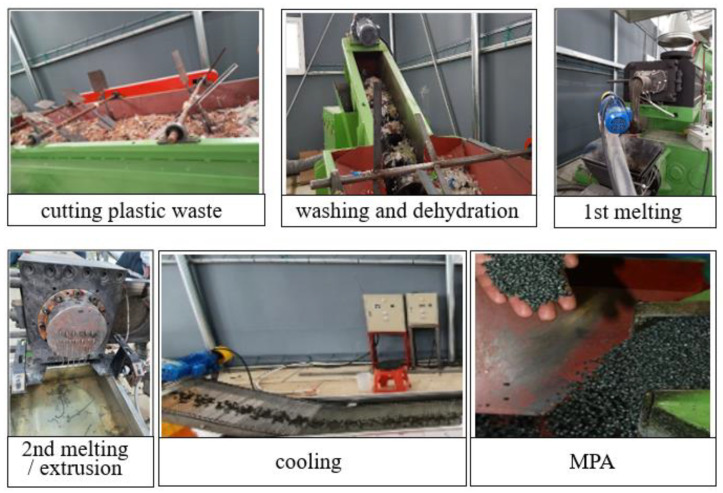
Manufacturing process of mixed plastic waste aggregates (MPAs).

**Figure 2 materials-17-00247-f002:**
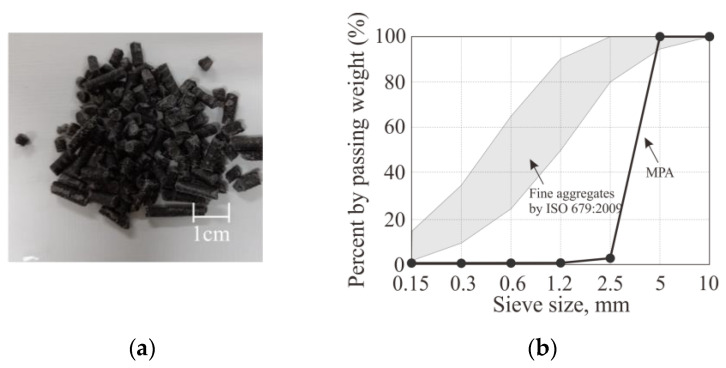
Particle size and shape of MPA: (**a**) shape and (**b**) particle size distribution [19].

**Figure 3 materials-17-00247-f003:**
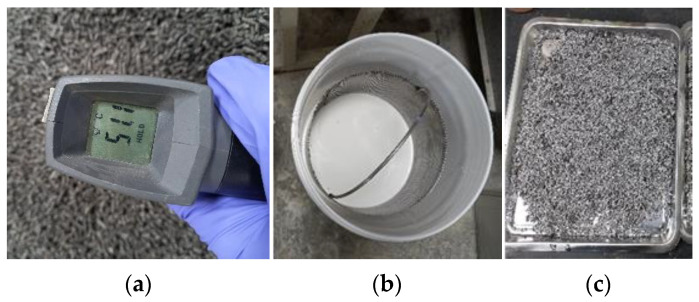
Procedure of surface hydrophilization: (**a**) heating, (**b**) impregnation, and (**c**) drying.

**Figure 4 materials-17-00247-f004:**
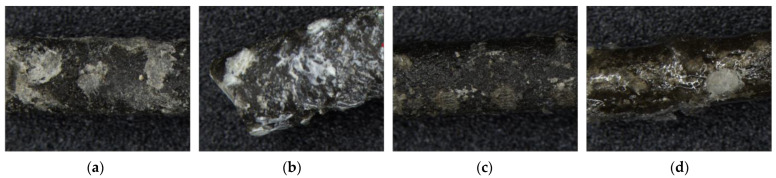
Surface condition of MPA: (**a**) unmodified, (**b**) modified with CA1, (**c**) modified with CA2, and (**d**) modified with CA3.

**Figure 5 materials-17-00247-f005:**
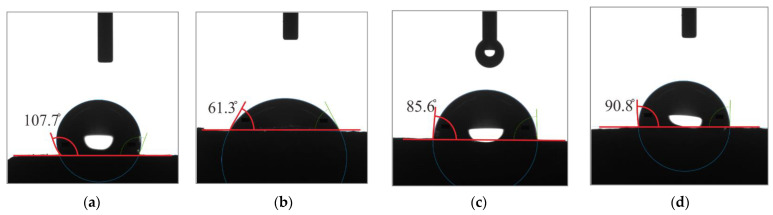
Contact angle of MPA: (**a**) unmodified, (**b**) modified with CA1, (**c**) modified with CA2, and (**d**) modified with CA3.

**Figure 6 materials-17-00247-f006:**
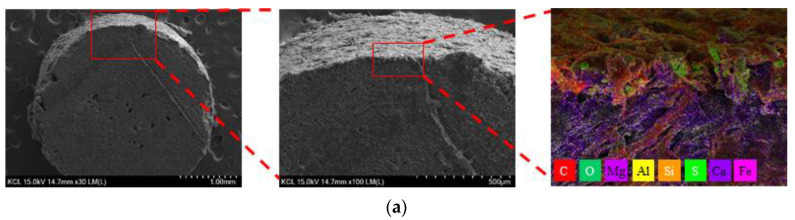
Scanning electron microscopy (SEM) images of MPA: (**a**) unmodified, (**b**) modified with CA1, (**c**) modified with CA2, and (**d**) modified with CA3.

**Figure 7 materials-17-00247-f007:**
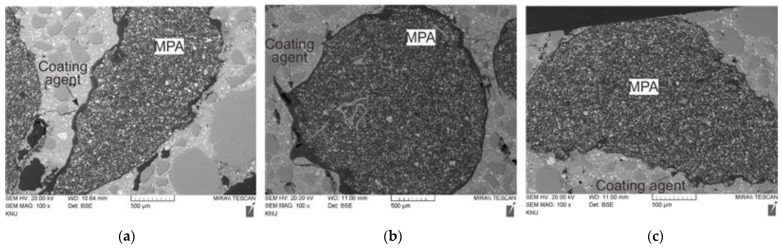
SEM images of mortar: (**a**) mix C1_MPA, (**b**) mix C2_MPA, and (**c**) mix C3_MPA.

**Figure 8 materials-17-00247-f008:**
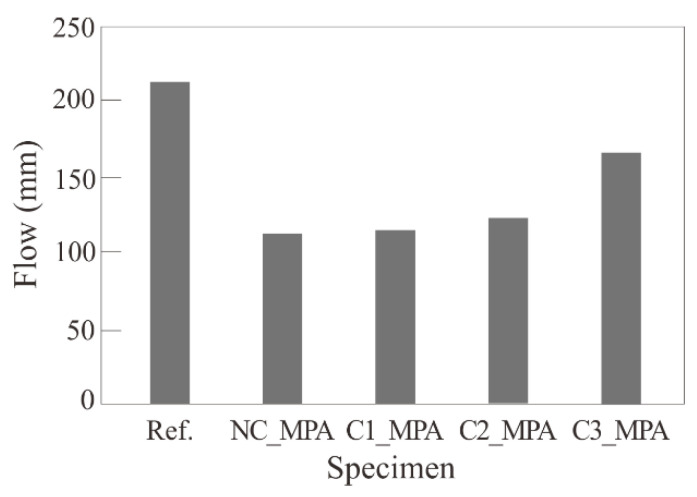
Flow by the surface modification of MPA.

**Figure 9 materials-17-00247-f009:**
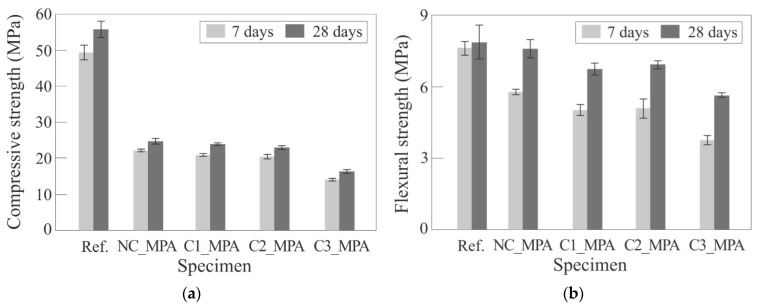
Strength by the surface modification of MPA: (**a**) compressive strength and (**b**) flexural strength.

**Figure 10 materials-17-00247-f010:**
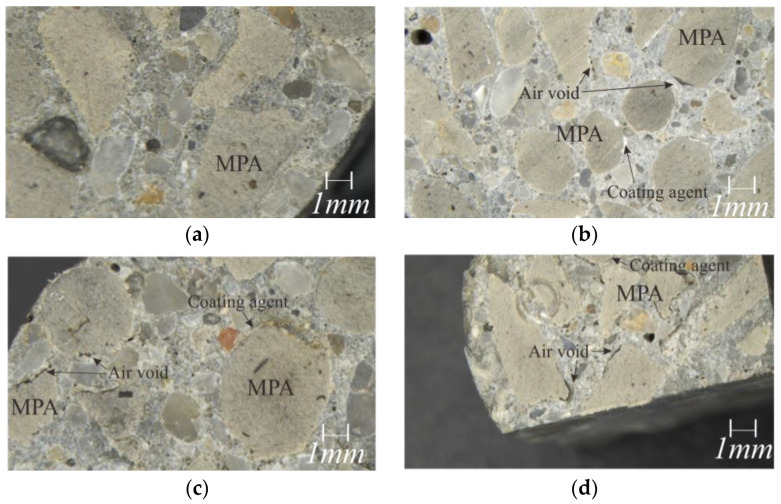
Optical microscope image of mortar: (**a**) mix NC_MPA, (**b**) mix C1_MPA, (**c**) mix C2_MPA, and (**d**) mix C3_MPA.

**Figure 11 materials-17-00247-f011:**
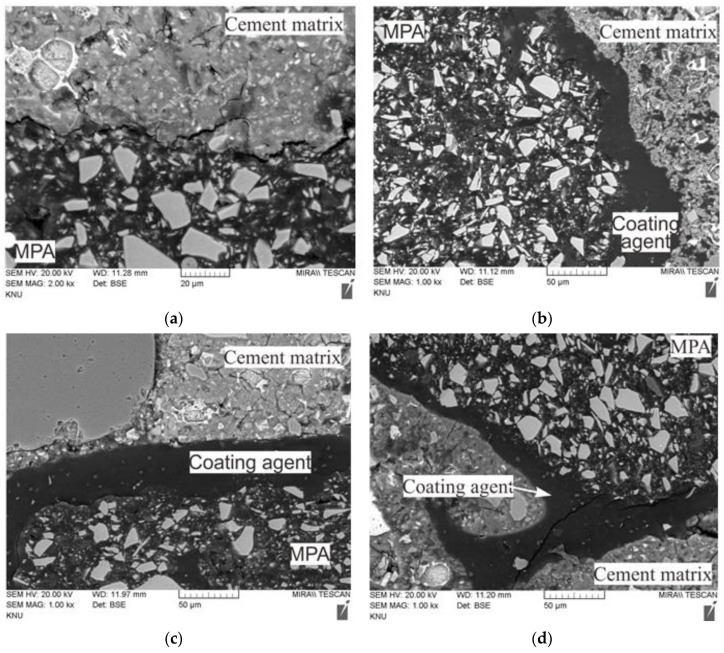
SEM images of mortar: (**a**) mix NC_MPA, (**b**) mix C1_MPA, (**c**) mix C2_MPA, and (**d**) mix C3_MPA.

**Table 1 materials-17-00247-t001:** Chemical composition of cement.

SiO_2_ (%)	Al_2_O_3_ (%)	Fe_2_O_3_ (%)	CaO (%)	MgO (%)	SO_3_ (%)
19.3	4.71	2.96	61.8	3.74	2.53

**Table 2 materials-17-00247-t002:** Physical properties of sand.

Fineness Modulus	Absorption Rate (%)	Max Size (mm)	Density (g/cm^3^)
2.83	2.20	2	2.48

**Table 3 materials-17-00247-t003:** Polymer composition of plastic waste.

Date	Polymer (%)					
	PE (LLDPE, LDPE, HDPE)	PP	PET	EVA	PVC	Others
23 May 2019	28	20	10	14	8	20
19 July 2019	42	18	8	12	4	16
27 August 2019	88	8	-	-	-	4
1 October 2019	68	12	4	8	-	8

**Table 4 materials-17-00247-t004:** Physical properties of MPA.

Fineness Modulus	Absorption Rate (%)	Density (g/cm^3^)
4.97	3.00	1.38

**Table 5 materials-17-00247-t005:** Mix design of mortar.

Mixture	Cement (kg/m^3^)	Water (kg/m^3^)	W/C (%)	Fine Aggregate (kg/m^3^)		Coating Agent
				Sand	MPA	
Ref.	450	171	38	1485	-	-
NC_MPA	450	171	38	647.5	599.7	-
C1_MPA	450	171	38	647.5	599.7	CA1 (silicone amine resin)
C2_MPA	450	171	38	647.5	599.7	CA2 (acrylic colorant)
C3_MPA	450	171	38	647.5	599.7	CA3 (ethyl acrylate binder, 10 wt%)

**Table 6 materials-17-00247-t006:** Contact angles of MPA.

	Contact Angle (°)							
Coating Agent	-		CA1		CA2		CA3	
Measurement Direction	Left	Right	Left	Right	Left	Right	Left	Right
No. 1	108.6	107.9	55.1	50.1	90.8	89.3	81.0	83.6
No. 2	100.9	101.4	62.5	67.6	86.0	87.7	92.4	90.6
No. 3	110.1	108.6	55.9	54.7	86.2	85.1	96.4	96.0
No. 4	97.4	94.7	61.0	62.2	79.0	82.9	91.8	89.8
No. 5	120.2	120.5	63.1	66.5	81.8	83.6	88.6	90.3
No. 6	111.4	110.4	67.0	69.8	86.5	88.1	93.8	95.2
Average/SD	107.7/8.03		61.3/6.13		85.6/2.97		90.8/4.71	

**Table 7 materials-17-00247-t007:** Distribution of elements constituting MPA and coating agents.

Coating Agent	Element (%)								Total
	C	O	Mg	Al	Si	S	Ca	Fe	
Uncoated	55.93	19.73	0.45	3.04	4.96	0.61	12.57	2.71	100
CA1	54.86	18.09	0.59	2.92	6.52	0.85	16.18	-	100
CA2	70.89	15.40	0.38	1.68	3.65	0.55	7.45	-	100
CA3	65.11	17.69	0.46	2.20	4.77	0.73	9.05	-	100

**Table 8 materials-17-00247-t008:** Flow and strength test results.

Mixture	Flow (mm)	Compressive Strength (MPa)				Flexural Strength (MPa)				Loss of Strength at 28 Days (%)	
		7 Days		28 Days		7 Days		28 Days		Compression	Flexure
		Average	SD	Average	SD	Average	SD	Average	SD		
Ref.	213	49.62	1.98	56.00	2.22	7.61	0.29	7.86	0.72	0	0
NC_MPA	113	22.10	0.36	24.68	0.72	5.79	0.10	7.58	0.40	56	4
C1_MPA	115	21.05	0.42	24.09	0.41	5.04	0.23	6.73	0.24	57	14
C2_MPA	122	20.35	0.67	22.99	0.40	5.09	0.40	6.92	0.17	59	12
C3_MPA	166	14.20	0.35	16.35	0.54	3.76	0.20	5.64	0.09	71	28

## Data Availability

Data is contained within the article.

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
