# Peer review of "Effect of Surface Hydrophilized Plastic Waste Aggregates Made by Mixing Various Kinds of Plastic on Mechanical Properties of Mortar"

_materials, 2024, doi:10.3390/ma17010247_

Round 1

Reviewer 1 Report

Comments and Suggestions for Authors

The review entitled "Effect of Surface Hydrophilized Plastic Waste Aggregates Made by Mixing Various Kinds of Plastic on Mechanical Properties of Mortar” improved the bond between MPA and the surrounding cement matrix using two types of coating agents. Specimens were prepared and tested under compression and flexure.

This manuscript is not well-written and is not well-prepared. It lacks clarity and it cannot be accepted in the current format. The following comments must be addressed.

Comments:

1-     Abstract: The methodology used in this investigation should be highlighted in the revised manuscript. Moreover, the results and conclusion should be concise and only the main conclusion should only highlighted.

2-     Introduction: Most of the literature is talking about concrete. However, the manuscript's title is about mortar. The presented figure in the introduction section is also talking about concrete.

3-     Figure 1: No need for this figure. The authors need only to report the previous research work in the text.

4-     Section 2.1: The term "Mixed plastic aggregates" should be consistent throughout the manuscript because the authors sometimes used "Mixed plastic waste aggregates".

5-     The authors still use the term "concrete" in the experimental work of this manuscript. However, the title was about mortar. More clarification should be provided. Based on the size distribution presented in Figure 3, this size of MPA is for concrete, not mortar.

6-     Page 3: Unclear sentence!

7-     Table 2 was not included in the text.

8-     Section 3.1: It should be "An ordinary", not "A ordinary".

9-     The size of the used MPA did not match the replaced sand. The size of MPA ranged between 2.5 mm to 5.0 mm. However, the maximum size of the sand was 2.0 mm only.

10- Section 3.2: This reviewer got confused. Was it concrete or mortar?

11- The specimen size of 40x40x160 for which type of tests? Were they for flexural or compressive strength tests?

12- Section 3.3: I think workability is more relevant to concrete mixes, not mortar. I think the initial and final setting times are more appropriate.

13- Section 3.3: This reviewer thinks that this loading rate of 2400 N/second is very high and fast.

14- Section 4.3: The authors should provide more discussions about the reasons behind these results, not just saying which one is bigger and which one decreases.

15- Section 4.4: Unclear discussions about Figure 10 are not enough. The authors should provide the story behind this figure.

16- The authors should provide number lines to easily track the revised manuscript. Also, the authors should respond point-by-point for each comment and provide the number lines where to find the answer of the modifications.

Comments on the Quality of English Language

This manuscript is not well-written and is not well-prepared. It lacks clarity and it cannot be accepted in the current format.

Author Response

1) We appreciate the comment, and fully agreed with your opinion. We revised the abstract focusing on the main conclusions and highlights, as you recommended.  : Page 1, line 11-21

2) We appreciate the comment. We fully agreed with your opinion. Actually, most of the existing studies what we mentioned in the introduction section had replaced fine aggregates in concrete with plastic waste aggregates (PWA), and the bond property between PWA and surrounding cement matrix has been reported to influence the physical properties of concrete. The PWA targeted in this study is also of a size corresponding to typical fine aggregates and we applied PWA to mortar before concrete application to evaluate the bond property between PWA and surrounding cement matrix.

3) We appreciate the comment. We agreed with your opinion and deleted Figure 1. 

4) We appreciate the comment. The term "Mixed plastic aggregates” revised as "mixed plastic waste aggregates". : Page 3, line 110

5) 

We appreciate the comment. We revised the term "concrete” which is directly related to the experimental work except for the term "concrete” which is the final application target of mixed plastic waste aggregates (MPA) in this study such as “ ~ because we intend to ultimately apply MPA to the concrete of heavy civil structures” in section 2.1.2 (Page 4, line 124).

The maximum particle size of ISO standard sand is 2 mm as you mentioned in the other comment. Meanwhile, the maximum particle size of fine aggregates is 4.75 mm according to “ISO 679:2009 Cement – Test methods – Determination of strength”, and the particle size distribution of fine aggregates according to ISO 679 was added to Figure 2 as below. We used MPA to replace fine aggregates, and the MPA particles used in this study distributed between 2.5 and 5 mm. We also used ISO standard sand as a type of fine aggregate for comparison. Sentences in section 2.1.1 were added to enhance the description of these things. : Page 3, line 133-137 : Page 4, line 142-144

6) We appreciate the comment. We extensively revised page 3 and rewrite the text as follows in the order of the polymer composition, manufacturing, and physical properties of MPA. : Page 3, line 111-137

7) We appreciate the comment. Table 2 was added in the text with rewriting page 3 by the response for 6.   : Page 3, line 127

8) We appreciate the comment. It was revised as “An ordinary”. : Page 2, line 103

9)  We appreciate the comment. The maximum particle size of ISO standard sand is 2 mm as you mentioned. Meanwhile, the maximum particle size of fine aggregates is 4.75 mm according to “ISO 679:2009 Cement – Test methods – Determination of strength”. We used MPA to replace fine aggregates, and the MPA particles used in this study distributed between 2.5 and 5 mm. We also used ISO standard sand as a type of fine aggregate for comparison.

10) We appreciate the comment. We conducted the experiment with mortar.  

11) We appreciate the comment. We casted the specimens of 40x40x160 and conducted both of flexural and compressive strength tests according to ISO 679. However, the number of reference was missing and incorrect. Therefore, we have revised the reference number and added the explanation of test methods. : Page 6, line 187-189 : Page 6, line 203-206

12) We appreciate the comment. We agreed with your opinion because the initial and final setting times are conducted by collecting mortar samples excluding the coarse aggregate in the concrete. Based on the results of this study, when applying MPA to concrete in the future, we will also evaluate the initial setting and setting time for concrete to which MPA has been applied.

13) We appreciate the comment. We conducted the compressive strength test with the loading rate of 2400 N/s according to ISO 679.

14) We appreciate the comment. We reviewed and extensively rewrote the mechanical test results and discussions, including comparisons with references. : Page 10, line 284-304

15) We appreciate the comment. We used Figure 10 to explain the air void, and related information was described in the text as in comment 14 above. : Page 10, line 291-295

16) We appreciate the comment. We have provide the number lines in the revised manuscript and written the corresponding number lines with the answer for each comment.

Reviewer 2 Report

Comments and Suggestions for Authors

The manuscript entitled “Effect of Surface Hydrophilized Plastic Waste Aggregates Made by Mixing Various Kinds of Plastic on Mechanical Properties of Mortar” has been reviewed. The results are helpful. However, the manuscript needs to be well improved before acceptance. Detailed comments are as follows:

1.      “As of 2015, ……”, the data was too old. Please update.

2.      The structuring of the manuscript is confusing and should be rearranged. Section 2 should be moved to Section 3 Materials and Methods with the sequence of 2.1 Materials, 2.2 Preparation and 2.3 Method.

3.      In Table 3, the unit % is missing.

4.      In Table 6, average should be Average. In addition, standard deviations should be added in Average.

5.      In Table 7, wt% of element should be Element (%).

6.      In all tables, please more attention of the capitalization of first letters for column or line names.

7.      In Table 8, standard deviations should be included.

8.      In Figs. 8 and 9, error bars should be added.

9.      In Fig. 10, scale bars should be included.

10.  References should be revised as per guide for authors of Materials and checked items by items. Pay more attention to the following errors:

1)        The abbreviation of journal names (e.g., Ref. 1).

2)        DOIs for all journal articles (e.g., Ref. 3).

3)        The full capitalization of article names (e.g., Ref. 9).

4)        The capitalization of first letters in journal names (e.g., Ref. 7).

5)        The correct format of book or book sections (e.g., Ref. 6).

Comments on the Quality of English Language

The manuscript entitled “Effect of Surface Hydrophilized Plastic Waste Aggregates Made by Mixing Various Kinds of Plastic on Mechanical Properties of Mortar” has been reviewed. The results are helpful. However, the manuscript needs to be well improved before acceptance. Detailed comments are as follows:

1.      “As of 2015, ……”, the data was too old. Please update.

2.      The structuring of the manuscript is confusing and should be rearranged. Section 2 should be moved to Section 3 Materials and Methods with the sequence of 2.1 Materials, 2.2 Preparation and 2.3 Method.

3.      In Table 3, the unit % is missing.

4.      In Table 6, average should be Average. In addition, standard deviations should be added in Average.

5.      In Table 7, wt% of element should be Element (%).

6.      In all tables, please more attention of the capitalization of first letters for column or line names.

7.      In Table 8, standard deviations should be included.

8.      In Figs. 8 and 9, error bars should be added.

9.      In Fig. 10, scale bars should be included.

10.  References should be revised as per guide for authors of Materials and checked items by items. Pay more attention to the following errors:

1)        The abbreviation of journal names (e.g., Ref. 1).

2)        DOIs for all journal articles (e.g., Ref. 3).

3)        The full capitalization of article names (e.g., Ref. 9).

4)        The capitalization of first letters in journal names (e.g., Ref. 7).

5)        The correct format of book or book sections (e.g., Ref. 6).

Author Response

1) We appreciate the comment. The introduction section of "As of 2015, “ was revised using 2019 data reflecting what you mentioned. : Page 1, line 26-31

2) We appreciate the comment, and agreed with your opinion. We rearranged section 2 and 3. : Page 2-6, line 100-211

3) We appreciate the comment. The unit % in Table 1 was added. : Page 3, line 107

4) We appreciate the comment. "average” in Table 6 was revised to "Average", and the standard deviations in average was added as you mentioned. : Page 7, line 229

5) We appreciate the comment. It was revised as “Element (%)”. : Page 9, line 265

6) We appreciate the comment. We reviewed and corrected the errors of first letters including Table 6, 7 and 8.

7) We appreciate the comment. The standard deviations of strength test results in Table 8 was added. : Page 9-10, line 280

8) We appreciate the comment. The flow test was performed only once for each mixture, and the values in the graph of Figure 8 are measurement values, not average values. Meanwhile, we added error bar in Figure 9. : Page 11, line 314-317

9) We appreciate the comment. We added scale bar in Figure 10. :  Page 11, line 318-323

10) We appreciate the comment. We extensively reviewed and revised references according to the guide of Materials including what you mentioned. : Page 13-14, line 386-436

* Detailed response is provided in the uploaded file.

Reviewer 3 Report

Comments and Suggestions for Authors

The submitted Article with the Manuscript ID “materials-2776863” and the title “Effect of Surface Hydrophilized Plastic Waste Aggregates Made by Mixing Various Kinds of Plastic on Mechanical Properties of Mortar” presents an experimental study that investigates the recycled plastic waste discharged with other types of household waste as sand replacement in cement-based mortar. The surface hydrophilization of Mixed Plastic waste Aggregates (MPA) is examined to improve the bond conditions between MPA and the mortar using two coating agents: a silicone amine resin and acrylic binders. The authors have researched an interesting issue. This material could be part of the basis material for a future article. However, this version of the paper is not acceptable to be published due to the following:

The present text needs the essential structure and contents of a scientific article. The present text could be considered closer to a professional report than a research paper. Further, the experimental part is relatively limited and briefly described, while the test results seem confusing in several parts. For example, the improvement of the bond properties of the examined mortar is not apparent from the short findings reported in sub-section 4.4. Furthermore, explaining the materials and methods in detail requires an explanation and justification of the materials and methods. Moreover, the manuscript lacks clarity about several of the research project issues: general context and boundaries, research gap, and the contribution and novelty within the research field.

The authors are encouraged to include a more detailed commentary of the results and critically discuss the observations from this investigation with the existing literature since tests are merely described.

The manuscript also has flaws concerning the overall quality, including the value of contribution, methodology, and presentation style, which are low for a typical journal paper. The paper does not bring any significant new scholarly contributions or offer any noticeable improvement. The findings of this article are questionable.

Based on the above comments, it is suggested that the paper be rejected. However, based on the above recommendations, an extensively revised version of this paper is most welcome.

Author Response

1) 

We appreciate the comment. The highlights of this study are as follow;

  • This study discussed the surface hydrophilization of MPA and its effects on the mortar properties.
  • 34% of sand volume was replaced with MPA and the density of MPA was increased by adding BFS fine power.
  • The workability of the mortar mixes increased up to 1.47 times by the surface hydrophilic MPA.
  • The compressive and flexural strengths of the mortar mixes decreased using the surface hydrophilic MPA.
  • The physical bond was formed on the MPA surface by the surface hydrophilization, and the mortar properties were affected depending on the formation state.

We reviewed and extensively revised the text to make it appropriate to the format of the research paper including the structure, logic, explanation, expression, and so on reflecting what you mentioned.

2) We appreciate the comment. We extensively revised the text in section 3. Results and discussion and also added the discussion of our experimental results with references reflecting what you mentioned as follows. Accordingly, we added several references, too. : Page 6-12 line 212-341

3) We appreciate the comment. We extensively revised the text reflecting what you mentioned and the response for this comment was also included in the response for the 1st comment. Moreover, plastic waste aggregates have a lot of potential to solve the environmental problems of plastic wastes and construction industries by mass recycling of plastic waste and reducing consumption of limited natural resources. We believe that our study can contribute to mass recycling of plastic waste by applying various types of structures. 

* Detailed response is provided in the uploaded file.

Reviewer 4 Report

Comments and Suggestions for Authors

This manuscript investigated the influence of surface hydrophilization of plastic waste aggregates (MPA) on the bond properties between MPA and the surrounding cement matrix. The study is interesting. But some issues should be addressed at first.

1.      In the abstract, “the strength compressive” should be changed to “the compressive strength”.

2.      In Fig.3, it seems that the size of all the MPA particles is between 2.5 mm and 5 mm. Please explain the influence of the poor grading on the fluidity of cement mortar.

3.      It is better to use specific surface area to replace granulation rate in section 3.1.

4.      The title of Table 8 is “Test results”. It is inappropriate.

5.      The MPA addition resulted in a big decrease on the compressive strength of cement composites in Table 8 and Fig.9. How about the feasibility of application of MPA in concrete?

6.      It is recommended to list the conclusions one by one.

7.      Why not use FT-IR analysis to determine the hydration between WPA and cement?

Comments on the Quality of English Language

none 

Author Response

1) We appreciate the comment. It was revised as “the compressive”. : Page 1, line 19

2) We appreciate the comment. We thought the particle distribution affected the decrease of the workability and the related information was added in the text. : Page 9, line 270-273

3) We appreciate the comment. We revised “granulation rate” to “specific surface area”. : Page 2, line 103-104

4) We appreciate the comment. The title of Table 8 was revised to  “Table 8. Flow and strength test results.". : Page 9, line 280

5) We appreciate the comment. Even though the compressive strength decreases due to the replacement of natural aggregates with MPA. However, we believed that the mechanical properties can be adjusted depending on the purpose of use of MPA, such as by adjusting the input amount of MPA. Moreover, MPA has a lot of potential to solve the environmental problems of plastic waste and construction industries. The related sentence was added in section 4. : Page 13 line 367-374

6) We appreciate the comment. The conclusion section was revised to write conclusions one by one as follows reflecting what you mentioned. : Page 12-13 line 343-374

7) We appreciate the comment. We understand a FT-IR analysis is useful for detecting the functional groups. In this study, we evaluated the surface hydrophilization of MPA using a contact angle and also evaluated whether the coating layer was formed or not and the distribution of elements constituting MPA and coating agent using scanning electron microscopy (SEM)-energy dispersive X-ray spectroscopy (EDS) analysis. In future research, we will also consider the application of the FT-IR analysis for evaluating the surface hydrophilization.

* Detailed response is provided in the uploaded file.

Round 2

Reviewer 1 Report

Comments and Suggestions for Authors

The authors addressed most of the reviewer's comments.

Reviewer 2 Report

Comments and Suggestions for Authors

The manuscript has been well revised. It can be accepted now.